# Finite Element Analysis of Ocular Impact Forces and Potential Complications in Pickleball-Related Eye Injuries

**DOI:** 10.3390/bioengineering12060570

**Published:** 2025-05-26

**Authors:** Cezary Rydz, Jose A. Colmenarez, Kourosh Shahraki, Pengfei Dong, Linxia Gu, Donny W. Suh

**Affiliations:** 1Gavin Herbert Eye Institute, Department of Ophthalmology, University of California, Irvine, CA 92617, USA; cezryd@gmail.com (C.R.);; 2Department of Biomedical Engineering and Science, Florida Institute of Technology, Melbourne, FL 32901, USA; jcolmenarezm2022@my.fit.edu (J.A.C.);

**Keywords:** sports-related injury, ocular trauma, retinal injury, glaucoma, finite element analysis

## Abstract

**Purpose:** Pickleball, the fastest-growing sport in the United States, has seen a rapid increase in participation across all age groups, particularly among older adults. However, the sport introduces specific risks for ocular injuries due to the unique dynamics of gameplay and the physical properties of the pickleball. This study aims to explore the mechanisms of pickleball-related eye injuries, utilizing finite element modeling (FEM) to simulate ocular trauma and better understand injury mechanisms. **Methods:** A multi-modal approach was employed to investigate pickleball-related ocular injuries. Finite element modeling (FEM) was used to simulate blunt trauma to the eye caused by a pickleball. The FEM incorporated detailed anatomical models of the periorbital structures, cornea, sclera, and vitreous body, using hyperelastic material properties derived from experimental data. The simulations evaluated various impact scenarios, including changes in ball velocity, angle of impact, and material stiffness, to determine the stress distribution, peak strain, and deformation in ocular structures. The FEM outputs were correlated with clinical findings to validate the injury mechanisms. **Results:** The FE analysis revealed that the rigid, hard-plastic construction of a pickleball results in concentrated stress and strain transfer to ocular structures upon impact. At velocities exceeding 30 mph, simulations showed significant corneal deformation, with peak stresses localized at the limbus and anterior sclera. Moreover, our results show a significant stress applied to lens zonules (as high as 0.35 MPa), leading to potential lens dislocation. Posterior segment deformation was also observed, with high strain levels in the retina and vitreous, consistent with clinical observations of retinal tears and vitreous hemorrhage. Validation against reported injuries confirmed the model’s accuracy in predicting both mild injuries (e.g., corneal abrasions) and severe outcomes (e.g., hyphema, globe rupture). **Conclusions:** Finite element analysis provides critical insights into the biomechanical mechanisms underlying pickleball-related ocular injuries. The findings underscore the need for preventive measures, particularly among older adults, who exhibit age-related vulnerabilities. Education on the importance of wearing protective eyewear and optimizing game rules to minimize high-risk scenarios, such as close-range volleys, is essential. Further refinement of the FEM, including parametric studies and integration of protective eyewear, can guide the development of safety standards and reduce the socio-economic burden of these injuries.

## 1. Introduction

Pickleball is the fastest-growing sport in the United States, and the number of participants is growing rapidly [1]. The number of players worldwide is increasingly associated with ocular injuries. The prevalence of injuries is higher in the elderly population, which constitutes a significant fraction of the growing player base. The sport has recently grown in popularity, attracting participants of all ages. However, it is important to highlight the specific risks to the eyes due to the fast pace of play and the nature of the ball. Eye injuries in pickleball can affect periocular tissue, anterior and posterior segments, and range from minor to severe. Previously reported ocular injuries associated with pickleball include partial vision loss, corneal abrasion, iritis, iris tear, hyphema, retinal tears, and vitreous hemorrhage [2,3,4]. These injuries and their pathophysiology are similar to those observed in other racquet sports like tennis, squash, or badminton where the eye is exposed to high-energy impact at high speeds, leading to significant trauma [5,6].

The pickleball used in the game is smaller and lighter than a tennis ball but more rigid, as it is made from hard plastic, making it stiffer and less compressible on impact. The inability to elastically store energy is precisely what makes blunt trauma a commonly observed mechanism of eye injury in pickleball accidents, as most of the kinetic energy is directly transferred to the eye. According to Pickleball Magazine, a pickleball can travel at approximately one-third the speed of a tennis ball, equating to around 40 miles per hour. The publication highlights that, when players are positioned at the “no-volley line”, the duration for a ball to traverse from one paddle to the other can range from 350 to 400 milliseconds. This rapid time frame allows little opportunity for players to evade a ball potentially striking them in the eye. The resulting trauma, such as orbital fractures or hyphema, can potentially lead to short- and long-term complications such as glaucoma. Interestingly, individuals with high myopia or a history of eye surgery are at an elevated risk for more severe consequences following pickleball-related ocular trauma.

The demographic profile of pickleball players further underlines the importance of protective measures. It is estimated that almost 20% of pickleball players are aged 65 and older, a group that, due to natural aging processes, do not possess the reflex and agility to avoid fast-moving objects. Furthermore, the prevalence of age-related ocular diseases such as cataract and age-related macular degeneration is high within this population, which can exacerbate the risk of long-term complications following the injury [5]. Given the rising popularity of pickleball, it is crucial to advocate for proper precautions and eye protection. According to the report by The American Academy of Ophthalmology, 30,000 sports-related eye injuries occur annually in the United States, many of which could be prevented with proper eye protection [5,6,7]. Proper advocacy is urgently needed for pickleball to prevent further rise in sports-related eye trauma.

Despite the risks and high speed of play, the use of protective eyewear in pickleball is very rare, leaving the participants of the sport vulnerable to injury. Although there are no large-cohort epidemiologic data, experts recommend wearing protective polycarbonate safety glasses [1,6,7,8,9]. Such eyewear can significantly reduce the likelihood of eye injuries by absorbing and deflecting the impact forces that would otherwise be transmitted to the eye. One study indicated that protective eyewear should be particularly considered for individuals at an elevated risk of retinal detachment. This includes those with a family history of the condition, individuals aged between 60 and 70 years, those with high myopia, or those who are pseudophakic [4].

The socioeconomic burden of pickleball-related injuries is substantial and is expected to rise in the coming year. Current estimates suggest that the total medical costs reached between USD 250 million and USD 500 million in 2023 alone [10]. These expenses, which include emergency room visits, outpatient treatments, surgeries, and hospitalizations, highlight the economic financial burden of injuries that could be significantly reduced with preventive measures (polycarbonate glasses).

In summary, as pickleball continues to gain popularity across all age groups and especially in the elderly population, the incidence of related eye injuries is expected to rise. Proactive steps need to be taken to promote the use of protective eyewear. It is of paramount importance to gain further understanding of the mechanics of these injuries to properly develop guidelines and recommendations.

In this study paper, we investigate the potential ocular trauma following pickleball-associated eye injury. Despite the sport’s growing popularity, there is no reliable research on these effects. Moreover, because of the sport’s recent rise in popularity, clinical case reports are also limited. Traditional research methods face limitations in accurately measuring forces applied on various eye structures within the eye. Complicated eye anatomy and tissue heterogenicity adds another layer of complexity. To overcome the above-mentioned challenges, we employed finite element (FE) computer simulation models. FE simulation has been successfully used in materials science. The computer simulation FE models allow for precise manipulation of various parameters (stress, energy, pressures, intraocular pressure, etc.), providing information about the biomechanics of pickleball-related eye injuries. FE models have been proven as an effective tool in studying ocular injuries across different sports and trauma mechanisms [11,12,13,14]. By applying this technology to pickleball, we aimed to perform a fluid–structure interaction analysis of how different segments of the eye respond to impact and its effect. We believe that the findings from this study will contribute to the development of evidence-based recommendations for eye protection for all pickleball players. Furthermore, we hope to influence the design and proper educational materials with the results of our research. Ultimately, the goal of our work is to improve our knowledge of the mechanism of pickleball-related eye trauma to raise awareness among eye care specialists, enhance player’s safety, and reduce the incidence of vision-threatening injuries.

## 2. Materials and Methods

### 2.1. Model Construction

A 3D model of an adult human eye embedded within a head was developed to simulate the biomechanical response to a pickleball impact (Figure 1A). Specific anterior intraocular structures such as the ciliary body, iris, and zonules were incorporated to assess the localized stress distribution, potential deformation, and risk of injury to these delicate tissues during impact. Zonules were modeled as a continuous material sheet, to capture the load transferring functions of the discrete fiber arrangement. The posterior ocular structures like the optic nerve head and macula were omitted based on the assumption that impact forces would primarily affect more superficial eye regions. Geometrical parameters were sourced from various studies: corneo-scleral thickness variations were based on measurements from Stitzel et al. [15]. The lens contour was constructed using polynomial coefficients provided by Burd et al. [16]. The thicknesses of the choroid and retina were chosen according to values reported by Thompson et al. [17]. The sagittal cross-section of the ciliary muscle was defined using dimensions from Knaus et al. [18], with its volume segmented into longitudinal, reticular, and circular fiber sections based on histology images from Tamm et al. [19]. Finally, four zonular divisions, represented as continuous annular sheets of material, were inserted between the apex of the ciliary muscle and the anterior, equatorial, and posterior regions of the lens. The corresponding thickness of these annular sheets was selected according to the zonular fiber diameter data obtained from Weeber and van der Heijde [20].

The head model was retrieved from our previous study [21], originally generated from the segmentation of high-resolution MRI images. The bony orbit of the eye was created by projecting the orbital rim dimensions adopted by Fitzhugh et al. [22] and creating a conical cutout 50 mm deep through the solid head model with a 45° angle with respect to the midsagittal plane. An inner canthal distance of 25 mm was selected for positioning the orbit cavity. Concurrently, the eye was situated at a neutral gaze angle in the center of the orbit, protruding 13 mm from the medial orbital rim. The medial, lateral, inferior, and superior rectus muscles were also included to fixate the eye within the orbit, with their insertion points defined by standard measurement of the spiral of Tillaux. The surrounding adipose tissue was assigned as the remaining geometry following the orbit cutout procedure and the removal of the space occupied by the eye and extraocular muscles. Once all the ocular structures were properly arranged, we converted the solid head into a surface body, as only the facial topology was of interest in the present work.

The pickleball ball was designed according to the regulatory dimensions set by the national governing body, USA Pickleball, with a specified diameter of 73 mm. To achieve the most rigid and impactful configuration of the ball, the minimum allowable number of circular holes, which is 26, was selected. The thickness of a pickleball depends on whether it is designed for indoor or outdoor play, with no regulatory specifications governing this aspect. Thus, we set a thickness of 1.8 mm based on self-made measurements on commercially available pickleball designed for outdoor games, as they are typically thicker.

### 2.2. Computational Simulation of Pickleball Impact

The blunt trauma sequela resulting from a pickleball impact was investigated using the finite element (FE) method. Specifically, the explicit FE solver from ABAQUS 2024 (Dassault Systemes Simulia Corp., Providence, RI, USA) was used to evaluate the dynamic response of the eye. A single impact simulation was conducted, representing a direct impact to the center of the cornea by a pickleball traveling at 40 miles per hour, which corresponds to the average serve speed in a casual game. The head surface was treated as a rigid and immutable body, since the skull stiffness significantly exceeds that of the ocular tissues, and the pickleball mass is insufficient to induce any head rotation. To accurately model the response of humor structures to impact forces, we represented the eye’s fluids as Eulerian elements and linked them to the solid structures using an Eulerian–Lagrangian coupling (ELC) approach. Due to modeling limitations at the ELC interface, vitreoretinal adhesion was assumed to be absent. This technique is especially useful in avoiding element distortions during large deformations and intense dynamic interactions, while efficiently capturing the fluid mechanics. On the other hand, contact interactions across the entire model were handled using a penalty method with a 0.5 friction coefficient. Only the zonular sheets were excluded from self- and fluid-contact interactions, as they consist of fibers that allow normal aqueous flow in the anterior chamber, rather than forming a continuous barrier. In total, 977,360 elements were used for the eye/head mesh, of which 216,000 corresponded to the fluid domain. The entire pickleball impact and rebound were simulated over a 0.005-s time frame, during which stress and deformation data were extracted. Gravity was also considered in the simulation as it influences the direction of the pickleball at the rebound.

### 2.3. Material Properties

The mechanical material properties for each ocular structure were adopted from the literature. Most of the soft tissues were modeled as hyperelastic, except for the zonules, which were treated as linear isotropic based on experimental observations [23]. Anisotropy was also included in those tissues whose response to different loading directions was available, e.g., lens capsule, and cornea. For the particular case of the ciliary body, we noticed a lack of specific mechanical information regarding the fiber’s contribution to its strength. Considering the pathological importance of its fiber orientation in subsequent blunt trauma injuries, we decided to fit the uniaxial data extracted from the smooth muscle of the carotid artery [24], to the Gasser–Holzapfel–Ogden (GHO) anisotropic model, while constraining the curve fitting procedure to a perfectly aligned fiber orientation (i.e., κ=0) in the direction of the muscle bundle. This approach was selected because the ciliary body shares functional and structural similarities with this type of smooth muscle. The remaining solid ocular structures were considered isotropic, with their hyperelastic material parameters either fitted from experimental results or analytically derived from Young’s modulus measurements. Humor fluids were modeled as nearly incompressible, viscous Newtonian fluids, with properties similar to water, as their water content is near 99%. Specifically, a Mie–Grüneisen equation of state was used as the material model, given the recommendations in the ABAQUS user’s guide. Finally, the pickleball was assigned as acrylonitrile butadiene styrene (ABS), which is linearly isotropic by nature. Specifications and references to the implemented material properties are shown in Table 1.

## 3. Results

### 3.1. Impact of Pickleball-Related Trauma on Anterior Segment of the Eye

The direct impact of a pickleball on the eye results in varying forces being applied to the structures of the anterior chamber. Interestingly, stress concentrations were observed in the iridocorneal angle region (Figure 2B), particularly in the area where the reticular muscle fibers of the ciliary body attach to the iris root. These stresses increase in intensity while the pickleball maintains full contact with the eye, reaching their peak at the point of maximum eye compression (t = 1.10 ms), just as the pickleball starts to bounce back. The primary factor contributing to the high stresses was a significant increase in the depth of the iridocorneal angle, caused by the pressure buildup in the anterior chamber. This pressure pushes the iris backward along with the inertial forces and stretches its junction with the ciliary body.

While examining the time evolution of iridocorneal average stress (Figure 2D), we noticed a rapid decrease as the eye returned to its original shape. Nonetheless, a secondary tensile peak occurred following the pickleball impact (t = 2.2 ms), as the anterior structures (i.e., sclera, cornea, ciliary body) pulled the lens forward to its original position, generating tensile forces at the ciliary body apex. Since the lens is suspended by thin zonule fibers, the ciliary–lens connection functions like a two-degree-of-freedom system, with the lens’s movement lagging behind the anterior structures. This delay in the lens’s rebound caused the ciliary body to tense the zonule fibers, resulting in strain at the zonular insertion zone. Thus, it is reasonable to believe that this secondary stress peak originating from the ciliary-to-lens motion interaction can further deepen the anterior chamber angle by extending any existing tear created upon impact.

Similar to ciliary body injury, the iris was primarily subjected to extensive tensile forces from pressure buildup and inertial motion. However, multiple secondary peaks were observed following the impact. This occurs due to the coup–contrecoup response of the humor fluid within the eye, which initially pushed the iris backward during the coup phase and then forward during the contrecoup phase, temporarily modifying the angle (Figure 2C). The oscillating motion of the iris continued until most of the dynamic energy was dissipated. This rapid fluttering of the iris caused the observed stress peaks at its root, which compromises the integrity of the tissue and could lead to possible iridodialysis.

The corneoscleral region where the trabecular meshwork is located also experienced significant stresses (Figure 2A). Initially, a minor compressive component was noted during the cornea’s flattening phase. After the cornea became fully flattened, a substantial tensile stress emerged as the eye was pushed backward. The maximum stresses occurred during this tensile stretching, marking the critical point for potential damage to the trabecular meshwork. These stresses acted circumferentially, as the limbus elongated due to the radial expansion of the cornea. Thereafter, the stress dropped, transitioning to a circumferential compression due to a sudden shrinking and buckling of the cornea upon pickleball bouncing. Noteworthy residual stresses were also noticed at the end of the simulation (0.35 MPa). Although we did not monitor the stress response over an extended period, residual stresses beyond physiological conditions likely exist in the limbal region, given that the stresses recorded were an order of magnitude greater than those in other parts of the eye. Additionally, the stress history suggests a steady, prolonged decay that will ultimately stabilize at a constant value. In tissues like the trabecular meshwork, residual stresses can trigger an inflammatory response by disruption of its structural integrity and homeostasis, potentially impacting the flow of the aqueous humor.

### 3.2. Dynamic Changes in Lens Zonules upon Impact

We closely followed changes in zonular tension to investigate the dynamics leading to a potential lens dislocation. Figure 3A displayed how the average stress for each zonular insertion varied over time. Upon impact, we observed zonular extension due to pressure increase in the anterior chamber. However, the secondary dynamic effects after the initial zonular extension marked the maximum stresses for the zonules. Given that both ciliary body and lens move close to each other during the initial compression phase of the eye, their slight relative displacement only generates the tensile stresses observed in the first peak (t = 0.8 ms). Nonetheless, it is the faster rebound of the anterior eye structures relative to the lens which causes a greater strain on the zonules (as explained in the previous section), severely increasing the potential for a tear. When comparing the stress at the different zonular insertions, we noticed that the equatorial zonules sustained the most damage (0.37 MPa), followed by the anterior (0.33 MPa), posterior (0.31 MPa), and anterior vitreous zonules (0.28 MPa). Additionally, markedly higher stresses were observed in the temporal region, showing a 34% difference compared to the zonules on the nasal side (Figure 3B). This observation is due to greater scleral bending on the temporal side during eye compression (Figure 3C), storing more elastic energy for release during the rebound phase, thereby pulling the temporal zonule fibers even more (Figure 3D). Since the temporal side of the eye is more exposed than the nasal, the observed bending asymmetry is expected. The nasal side benefits from better support from the surrounding orbital fat and bone, which helps prevent excessive flexural movement—a condition not present on the temporal side.

### 3.3. Pickleball-Related Injury Impact on the Retina

To investigate the dynamics underlying posterior segment damage, we analyzed the distribution of von Mises stress in the retina after impact. Our analysis revealed that the areas exposed to the highest stress included the ora serrata and the posterior pole (Figure 4A and Figure 4B, respectively). Interestingly, these regions coincided with the sites of maximum pressure buildup within the vitreous (Figure 4C,D), implying that retinal stresses following blunt trauma are directly related to the local increase in intraocular pressure (IOP) adjacent to the retina. This connection was further corroborated by the orientation of the maximum absolute principal stress on the retina, which was predominantly compressive because of the hydrostatic forces exerted by the vitreous. Although this major stress component was compressive, soft tissues rarely fail under compression. Therefore, upon comparing the remaining tensile and shear components, we found that shear stresses were significantly higher, underscoring their crucial role in the pathogenesis of retinal tears and breaks. It can be inferred that the concentration of IOP at the ora serrata is likely a key mechanism behind retinal dialysis, as the induced shear stress can disrupt the junction between the retina and the ciliary body.

Because of the crucial influence of IOP in retina biomechanics, we then analyzed its local time evolution in the ora serrata, posterior, nasal, and temporal regions (Figure 5). At the ora serrata, a rapid spike was observed, which quickly dropped once the impact wave travel back through the eye. Shortly after, the measured pressure in the nasal and temporal regions peaked, as the wavefront reached the eye’s equator. However, following this brief peak, the IOP in these regions did not drop as rapidly as at the ora serrata, instead maintaining an elevated level. Finally, when we examined the pressure levels at the posterior pole, we observed a slower buildup of pressure, followed by a delayed spike and then a gradual pressure decrease.

Retinal traction is also a critical factor to evaluate in ocular blunt trauma, as it can lead to the formation of retinal holes and subsequent detachments. Thus, we examined the fluid dynamics of the vitreous in search of any drastic changes in velocity or pressure gradients that could potentially contribute to tractional forces. At the onset of the contrecoup motion, the temporal side exhibited a higher localized pressure than the nasal side due to the accumulation of more vitreous material (Figure 6A). This asymmetry arose from the nasal side’s faster volumetric expansion, which led to a drop in pressure and vitreal density. Consequently, a greater opposite reaction from the vitreous–retina interaction occurred on the temporal side, as shown by the velocity field in Figure 6B. Since the vitreous is attached to the retina, we hypothesize that this intense momentum vector generated during the contrecoup can create substantial tractional forces, potentially resulting in retinal rupture. Although we did not consider any adhesive properties in the vitreoretinal interface, we visualized an isolated vitreous separation on the temporal side due to this reaction mechanism (Figure 6C).

## 4. Discussion

Pickleball, a rapidly growing sport in the United States, has seen a significant increase in participation, particularly among individuals aged 65 and older. This rise in popularity has resulted in an increase in pickleball-related ocular injuries. These injuries are typically less common than musculoskeletal injuries; however, they lead to severe and long-lasting vision impairment and disability. Understanding the underlying biomechanics of these injuries is crucial for healthcare providers to improve diagnostic and therapeutic protocols and for further development of evidence-based guidelines. As the sport continues to evolve, the meticulous tracking of injury types and mechanisms will be crucial for informing effective injury prevention strategies and enhancing the safety of players.

Previous clinical reports have documented several cases of eye injuries resulting from pickleball. For instance, a 47-year-old male suffered a hyphema and elevated intraocular pressure. Another group reported a 48-year-old male experienced subconjunctival hemorrhage and commotio retinae [34]. Other case reports include a 66-year-old male who sustained a posterior vitreous detachment and retinal tear, and a 77-year-old male who experienced intraocular lens dislocation [7,34]. Huang and Greven [3] presented two cases of traumatic lens subluxation resulting from pickleball injuries. Reported cases highlight the wide spectrum of ocular injuries that can result from participation in pickleball. Our finite element analyses align with previously reported clinical cases, providing biomechanical insight into the mechanisms underlying the diverse ocular injuries observed in pickleball-related trauma. Interestingly, our FEM identified peak zonular stresses (up to 0.37 MPa) during the rebound phase, which aligns with clinical reports of traumatic lens subluxation by Huang and Greven (2024) [3]. Their cases documented zonular fiber tears in patients after pickleball impacts, corroborating our finding that equatorial zonules are most vulnerable to dynamic forces.

The observed shear-stress concentrations at the ora serrata and posterior pole (Figure 4A,B) correlate with retinal tears reported by Atkinson et al. (2022) [4]. Their cases highlighted posterior pole tears following blunt trauma, consistent with our simulation’s prediction of elevated intraocular pressure (IOP) gradients disrupting retinal adhesion.

Stress peaks in the iridocorneal angle (0.35 MPa, Figure 2D) correspond to clinical observations of hyphema and angle recession by Waisberg et al. (2023) [7]. These injuries result from trabecular meshwork disruption, which our model attributes to circumferential tensile stresses during limbal expansion.

Asymmetric vitreous pressure gradients (Figure 6A–C) explain cases of vitreous hemorrhage and posterior vitreous detachment in Boopathiraj et al. (2024) [34], where temporal traction forces exceeded retinal resilience.

Given that a significant portion of pickleball players are aged 65 and older, there is a high probability of serious and debilitating ocular injuries [7]. In contrast to tennis, the majority of injuries encountered in pickleball predominantly affect older players rather than their younger counterparts. Notably, the incidence of pickleball-related injuries exhibits a positive correlation with advancing age. In our simulation, we did not account for the effects of aging. However, it is worth noting that the stresses and injuries discussed in this study are likely to be exacerbated by aging. With age, soft tissues undergo stiffening, and the vitreous body progressively liquefies and shrinks. These changes increase the susceptibility of ocular tissues, particularly at the vitreoretinal interface, making them more vulnerable to mechanical stress and injury. Previous biomechanical studies have demonstrated that aging significantly alters the mechanical properties of the eye. With advancing age, the crystalline lens becomes progressively stiffer due to protein aggregation and a reduction in elasticity [35]. In fact, lenses in older individuals can be up to four times stiffer compared to those in younger individuals [35]. Computational models have shown that increased lens stiffness leads to elevated peak stress in the ciliary body, thereby raising the risk of tissue rupture during blunt ocular trauma. Similarly, corneal stiffness increases with age and is associated with elevated intraocular pressure and a diminished capacity to withstand deformation, which heightens the risk of corneal abrasion and rupture upon impact [36]. Furthermore, age-related vitreous liquefaction reduces the eye’s ability to absorb mechanical shock, increasing the susceptibility to retinal detachment following trauma [37].

Despite the increasing incidence of pickleball-related eye injuries, there is a lack of comprehensive research on the specific risks associated with pickleball. We believe our results will expand our understanding and improve therapeutic outcomes. More studies are needed to understand the mechanisms of injury and to develop targeted prevention strategies.

The outcomes of our study offer valuable insight into the biomechanical factors involved in ocular injury caused by pickleball. By utilizing a finite element analysis model, we have gained a detailed understanding about how stress distribution and intraocular pressure change (IOP) affect the anterior segment, with lens and zonules in particular, and posterior segment. Our biomechanical evaluations are essential for clarifying the mechanism seen in pickleball-related ocular trauma.

The anterior ocular structures are particularly susceptible to biomechanical stresses, with the trabecular meshwork (TM) region experiencing the highest levels of stress, followed by the iridocorneal angle and the iris root. These stresses are primarily induced by the buildup of aqueous humor pressure, which exerts excessive tensile forces on the iridocorneal angle, disrupting the biomechanical equilibrium. High tractional shear stress resulting from pressure buildup may result in angle recession and hyphema and potentially secondary glaucoma as a long-term complication. These findings highlight the importance of thorough examination in patients after pickleball-related trauma due to the risk of long-term complications. Within the TM, significant residual stresses persist even after initial deformation events. This chronic stress state may act as a stimulus for aberrant cellular behaviors, such as proliferation, potentially contributing to pathological remodeling and dysfunction in outflow regulation.

It is important to distinguish between the primary (compression) and secondary (rebound) phases of ocular trauma, as each phase imparts unique mechanical forces that can differentially affect various ocular structures.

During the primary (compression) phase, a pickleball, traveling at speeds up to 40 mph, impacts the globe, causing rapid deformation at the point of contact. This compression phase, as previously analyzed with FEM, is characterized by a sudden increase in intraocular pressure and mechanical stress across both the anterior and posterior segments of the eye [38,39]. Notably, this impact affects various ocular structures in distinct ways. The cornea and sclera absorb the initial force, potentially resulting in corneal abrasions or, in severe cases, globe rupture [6,7]. In the compression phase, specific trauma in the iris can present as iridodialysis or sphincter muscle tears. At the ciliary body, posterior blunt trauma can lead to displacement of the iris root and lateral transmission of force via the aqueous humor. This may also result in hyphema if ciliary arteries are ruptured [39,40]. In the posterior segment, a rapid pressure wave travels through the vitreous, causing posterior displacement of the vitreous body. This can lead to vitreous detachment, retinal tears, or even localized retinal detachment, especially at sites where the vitreous is firmly adherent to the retina [1,6,7,38].

In the secondary (rebound) phase, the globe recoils and the intraocular pressure transiently drops below baseline. This can cause further injury as the ocular structures return to their original shape [38,40]. Within the iris and ciliary body, the rapid anterior movement can exacerbate tears or detachments initiated during compression. The ciliary body may suffer further from shearing forces, increasing the risk of angle recession or cyclodialysis cleft (separation of the ciliary body from the sclera) [40]. On rebound, the lens, if subluxated or dislocated during compression, may shift further, increasing the risk of traumatic lens dislocation or phacodonesis [7,39]. In the posterior segment, the rebound can cause additional traction on the retina, particularly if vitreous detachment has occurred, increasing the risk of retinal tears or detachment [1,6,7,38]. Secondary hemorrhage (vitreous or retinal) may also occur due to vascular shearing, as the vitreous body, having been displaced, may pull on the retina and contribute to further retinal pathology.

Given previously reported cases of lens dislocation following pickleball eye trauma, it was important to investigate the dynamic of changes within zonules. Maximum tensile stresses in the zonular fibers were observed during the rebound phase of pickleball impact, rather than during the initial compressive stage. This phenomenon is attributed to the faster recoil motion of the sclera and other anterior ocular structures compared to the lens. Interestingly, the equatorial zonules experienced the highest stress, underscoring their critical role in absorbing the dynamic forces generated by the rapid deformation and recovery of the eye’s anatomy. These findings partially explain the pathomechanism behind lens subluxation in pickleball eye trauma.

Moreover, we have observed differences in stress within the posterior segment compartment. Interestingly the highest observed stress involved the peripheral retina and ora serrata region. In our model, we observed that the retina is exposed to shear stresses that are directly linked to the local increase in the intraocular pressure, rather than deformations resulting from the impact to other structures (e.g., sclera). The maximum IOP peaks were fairly similar among all regions of the globe, confirming that both peripheral and posterior-central retina regions are equally vulnerable to injuries. Based on our results, we can infer that the adhesive properties of the retina at different locations are a primary factor in the development of a retinal tear. The analysis of the adhesive properties of the vitreoretinal interface will be a subject of further investigation to analyze tractional forces. The resulting stress in the ora serrata resulting from trauma may lead to retinal dialysis, a vision-threatening condition, commonly observed in ocular trauma cases [41,42,43]. In our simulation, we observed that pickleball can lead to stresses in the posterior retinal region, potentially leading to formation of retinal foramina or tears. Our findings are in line with previous reports of clinical cases with retinal tears in the posterior region following traumatic events [44,45,46]. Previous studies that included computational modeling of ocular trauma in different sports reported similar findings [38,47]. Our findings highlight the importance of a careful retinal exam in cases of pickleball-related eye injury.

The current lack of mandatory protective eyewear in pickleball is a significant oversight, especially considering the high risk of eye injuries. Protective eyewear, such as polycarbonate glasses, can significantly reduce the risk of injury by preventing direct contact with the eye [8]. Advocacy for eye protection in pickleball is essential, particularly those in higher-risk age groups. While our study offers a novel information into the biomechanical aspects of pickleball-related eye injuries through computational simulations, it is crucial to mention its limitations. The simulation was performed using previously reported stiffness and mechanical values [39,48]. These values often do not represent those seen in the aged eye. Moreover, due to the complexity of the human eye and the nature of sport, not all trauma mechanisms can be captured with computational models. Moreover, while valuable for simulating ocular biomechanics, finite element models may not fully capture the viscoelastic complexity of ocular tissues due to inherent simplifications in both geometry and constitutive behavior [49]. Geometric approximations often diverge from the true anatomical features of the cornea and sclera [50,51]. The accuracy and predictive power of these models depend on achieving an optimal balance between computational tractability and faithful representation of tissue microstructure and viscoelastic relaxation dynamics [49,51]. Our study did not include analysis of physiological response, vascular behavior, or long-term injury. Nevertheless, our data significantly increase our understanding of the forces involved in the pathomechanism of pickleball-related ocular trauma.

## 5. Conclusions

With the expected growth of the sport’s popularity, especially among older adults, it is important to raise awareness about the potential for eye injuries and the importance of protective measures. Treating sports injuries in geriatric patients is a relatively new topic for many ophthalmologists. Healthcare providers should educate patients on the risks and encourage the use of protective eyewear. Furthermore, sports organizations should consider implementing mandatory eyewear regulations to protect players.

In conclusion, the associated risk of eye injuries related to pickleball has to be further studied. Through increased research, awareness, and advocacy for protective measures, the pickleball community can work towards minimizing these risks and promoting a safer playing environment for all. Future studies will integrate polycarbonate lenses into our FEM to quantify stress reduction in ocular structures. This will inform ASTM/ISO standards for pickleball-specific eyewear, addressing the lack of regulatory guidelines noted by The American Academy of Ophthalmology (2023) [9]. Moreover, to specifically study risks in the aging population, a parametric study will model age-related changes (e.g., scleral stiffening, vitreous syneresis) to evaluate their compounding effects on injury risk. This is critical given that 20% of players are >65 years old, a cohort with heightened susceptibility to retinal detachment and zonular weakness. By addressing the biomechanical aspects of pickleball-related trauma, we aim to increase the understanding of the spectrum of mechanisms involved in pickleball-related eye injuries.

## Figures and Tables

**Figure 1 bioengineering-12-00570-f001:**
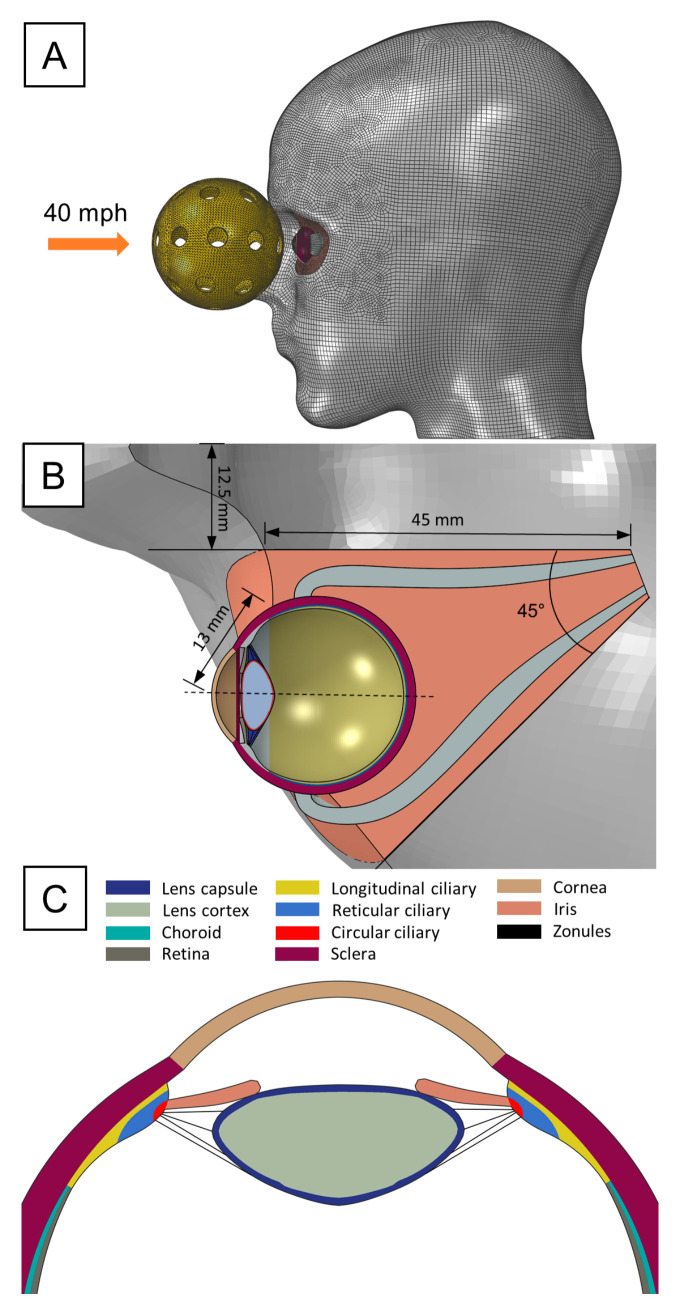
FE pickleball blunt trauma model: (**A**) Lateral view of the head showing initial pickleball’s position and employed mesh. (**B**) Axial view of the orbit and corresponding geometric specifications. (**C**) Transverse section of the anterior ocular structures.

**Figure 2 bioengineering-12-00570-f002:**
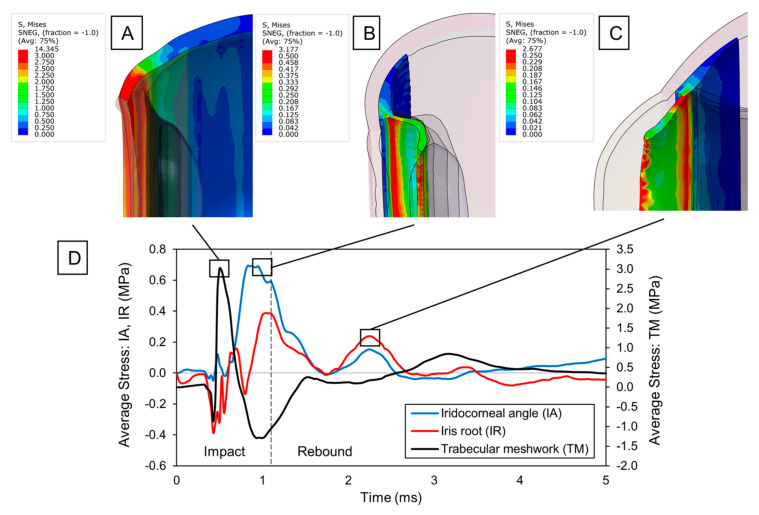
Blunt trauma to anterior ocular structures: (**A**) Von Mises stress distribution of the sclera upon max. stress in the trabecular meshwork. Stress concentrations are emphasized in the sclero-corneal junction. (**B**) Von Mises stress distribution of the ciliary body and iris upon max. stress at the iridocorneal angle. Stress concentrations are observed with increased depth in the iris angle. (**C**) Von Mises stress distribution of the ciliary body and iris upon secondary stress peak at the iridocorneal angle. Tensile stress arises due to contrecoup and inertial reactions in the iris root. (**D**) Time evolution of average stress at the iridocorneal angle (IA), iris root (IR), and trabecular meshwork (TM).

**Figure 3 bioengineering-12-00570-f003:**
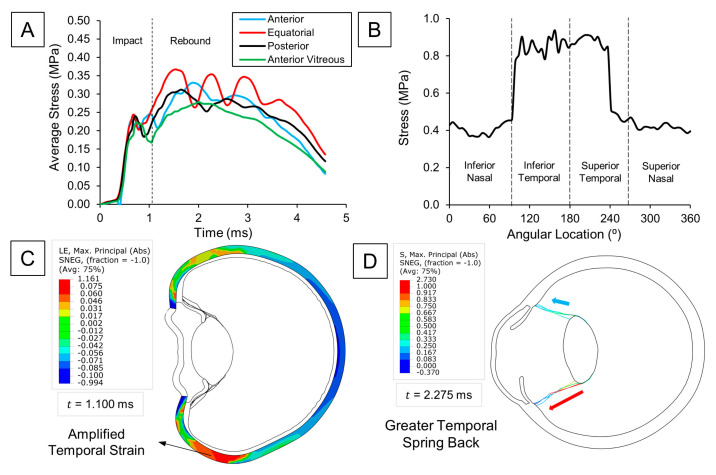
Zonular tension analysis: (**A**) Time evolution of average stress at the four zonular divisions. (**B**) Local variations in stress at the anterior zonules with respect to the angular location. (**C**) Max. principal strain (abs) distribution of the sclera. Higher strains are observed in the temporal side. (**D**) Max. principal stress (abs) of zonular fibers. Higher tensile stress is observed at the temporal side because of the greater strain energy release.

**Figure 4 bioengineering-12-00570-f004:**
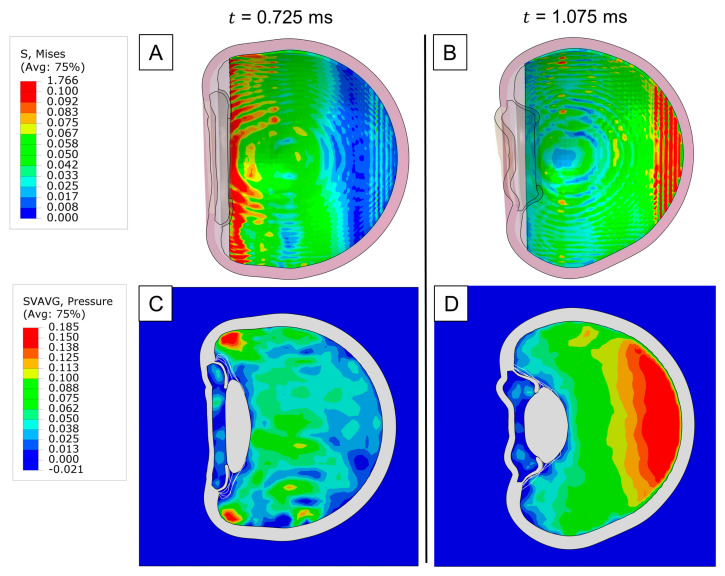
Impact injury on the retina: (**A**) Von Mises stress distribution of the retina upon max. stress at the ora serrata. (**B**) Von Mises stress distribution of the retina upon max. stress at the posterior pole. (**C**) Pressure distribution of the vitreous upon max. retinal stress at the ora serrata. (**D**) Pressure distribution of the vitreous upon max. retinal stress at the posterior pole.

**Figure 5 bioengineering-12-00570-f005:**
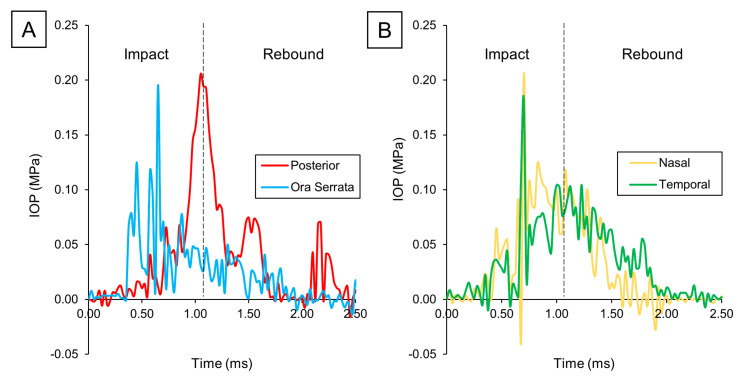
IOP variation over time: (**A**) Posterior vs. ora serrata. (**B**) Nasal vs. temporal.

**Figure 6 bioengineering-12-00570-f006:**
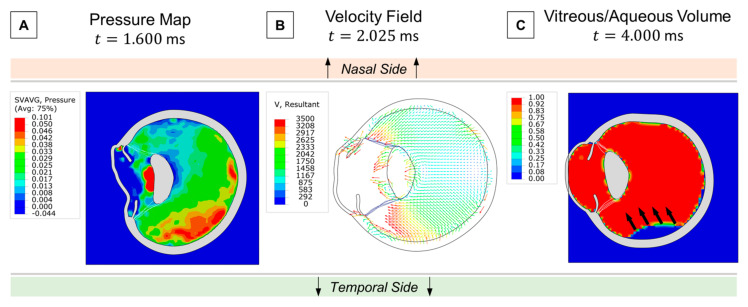
Retinal traction analysis: (**A**) Asymmetric pressure distribution of the vitreous at contrecoup. (**B**) Velocity field following temporal pressure buildup. (**C**) Vitreous separation from the retina by the end of the simulation.

**Table 1 bioengineering-12-00570-t001:** Summary of material properties for the ocular tissues. E and v represent the Young’s modulus and Poisson’s ratio, respectively. The material constants $c_{ij}$ for i,j=1,2,…,N define the strain energy density for the hyperelastic tissues.

Component	Constitutive Model	Material Constants	Reference
Pickleball	Linear elastic	E=2500 MPa ν=0.3	Li et al. [25]
Choroid	Neo-Hookean hyperelastic	$c_{10}=0.1\;\mathrm{MPa}$	Colmenarez et al. [26]
Ciliary body	GHO hyperelastic	$c_{10}=0.0493\;\mathrm{MPa}$ $k_{1}=0.006$ $k_{2}=0.0026$ κ=0	Herlihy et al. [24]
Cornea	GHO hyperelastic	$c_{10}=0.021\;\mathrm{MPa}$ $k_{1}=87.93$ $k_{2}=15.2$ κ=0.2	Nambiar et al. [27]
Zonules	Linear elastic	E=0.308 MPa ν=0.48	Michael et al. [23]
Lens cortex	Neo-Hookean hyperelastic	$c_{10}=0.05\;\mathrm{MPa}$	Weeber et al. [28]
Lens capsule	GHO hyperelastic	$c_{10}=0.075\;\mathrm{MPa}$ $k_{1}=6.93142$ $k_{2}=28.97$ κ=0.23	Berggren et al. [29]
Retina	Neo-Hookean hyperelastic	$c_{10}=0.05533\;\mathrm{MPa}$	Ciasca et al. [30]
Iris	Neo-Hookean hyperelastic	$c_{10}=0.1667\;\mathrm{MPa}$	Heys et al. [31]
Sclera	Yeoh’s hyperelastic	$c_{10}=0.91\;\mathrm{MPa}$ $c_{20}=19.023\;\mathrm{MPa}$ $c_{30}=-64.725\;\mathrm{MPa}$	Colmenarez et al. [32]
Orbital adipose	Neo-Hookean hyperelastic, and viscoelastic	$c_{10}=0.145055\;\mathrm{kPa},$ D=5 MPa	Chen et al. [33]
Humor bodies	Mie-Grüneisen equation of state with linear Us-Up Hugoniot condition	$c_{0}=1.45\times 10^{6}\;\mathrm{mm}/s$ s=0 $\Gamma_{0}=0$ $\eta =1\times 10^{-8}$ N·s/mm^2^	Dassault Systemes

## Data Availability

The original contributions presented in this study are included in the article. Further inquiries can be directed to the corresponding authors.

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
