# Peer review of "Finite Element Analysis of Ocular Impact Forces and Potential Complications in Pickleball-Related Eye Injuries"

_bioengineering, 2025, doi:10.3390/bioengineering12060570_

Round 1

Reviewer 1 Report

Comments and Suggestions for Authors

1. Explain in detail the sequence of mechanical events during the pickleball impact. especially, distinguish between primary (compression) and secondary (rebound) injury phases across various ocular structures such as iris, ciliary body, zonules, retina.

2. Being consistence with the following terms of for example "spring-back," "rebound," "contrecoup," and "compression phase." Some sections use these interchangeably, that is unclear for readers who unfamiliar with ocular biomechanics.

3.Stress values (e.g., 0.35 MPa, 0.37 MPa) are mentioned, mention the standard deviations if there are multiple simulations, or clarify if these are single-simulation results. Also, clarify if whether these stress levels exceed a specific value for the ocular tissues.

4. The paper demonstrated stress and pressure peaks over time. It may enhance clarity to explicitly specify time points or normalized time units when these peaks occur.

5. Add more detailed discussion on the modeling assumptions, particularly regarding: Whether vitreous liquefaction was considered, whether zonular fiber anisotropy was modeled, Simplifications at the vitreoretinal interface (no adhesion considered).). These assumptions will clarify the interpretation of the results.

6. Aging effects are mentioned briefly. Consider adding a short paragraph discussing how age-related changes in ocular biomechanic, which might influence the injury patterns observed in pickleball trauma.

7. Some figures (e.g., Fig. 2D, 3A, 5A) would benefit from clearer labeling of peaks and events (impact, rebound, etc.).

8. Please include scale bars and color legends in all stress and pressure distribution figures for better interpretability.

9. Where possible, draw stronger connections between simulation findings, and specific clinical case reports cited. this would strengthen the clinical relevance of the modeling results.

Finite element material models may not fully capture viscoelastic behavior of ocular tissues. Discuss that carefully.

12.Minor grammatical corrections needed throughout the text (e.g., "increase in participation" → "an increase in participation"; "potential of a tear" → "potential for a tear").

13.Consistently use "impact" instead of "hit" or "strike" unless describing an informal event.

14. Show the impact of your research as future work. 

Author Response

We sincerely thank the reviewer for taking the time to carefully read and evaluate our manuscript. Your thoughtful and constructive comments have greatly contributed to improving the clarity and quality of our paper on pickleball-related ocular injuries. 

Comment 1: Explain in detail the sequence of mechanical events during the pickleball impact. especially, distinguish between primary (compression) and secondary (rebound) injury phases across various ocular structures such as iris, ciliary body, zonules, retina.

Response to comment 1: Thank you for this suggestion. In response, we have incorporated a detailed explanation of the primary (compression) and secondary (rebound) phases into the Discussion section of the manuscript.

Line 454: “It is important to distinguish between the primary (compression) and secondary (rebound) phases of ocular trauma, as each phase imparts unique mechanical forces that can differentially affect various ocular structures.

During the primary (compression) phase, a pickleball, traveling at speeds up to 40 mph, impacts the globe, causing rapid deformation at the point of contact. This com-pression phase, as previously analyzed with FEM, is characterized by a sudden increase in intraocular pressure and mechanical stress across both the anterior and posterior segments of the eye. Notably, this impact affects various ocular structures in distinct ways. The cornea and sclera absorb the initial force, potentially resulting in corneal abrasions or, in severe cases, globe rupture. In the compression phase, specific trauma in the iris can present as iridodialysis or sphincter muscle tears. At the ciliary body, posterior blunt trauma can lead to displacement of the iris root and lateral transmission of force via the aqueous humor. This may also result in hyphema if ciliary arteries are ruptured. In the posterior segment, a rapid pressure wave travels through the vit-reous, causing posterior displacement of the vitreous body. This can lead to vitreous detachment, retinal tears, or even localized retinal detachment, especially at sites where the vitreous is firmly adherent to the retina.

In the secondary (rebound) phase, the globe recoils and the intraocular pressure transiently drops below baseline. This can cause further injury as the ocular structures return to their original shape. Within the iris and ciliary body, the rapid anterior movement can exacerbate tears or detachments initiated during compression. The ciliary body may suffer further from shearing forces, increasing the risk of angle recession or cyclodialysis cleft (separation of the ciliary body from the sclera). In rebound, the lens, if subluxated or dislocated during compression, may shift further, increasing the risk of traumatic lens dislocation or phacodonesis. In the posterior segment, the rebound can cause additional traction on the retina, particularly if vitreous detachment has occurred, increasing the risk of retinal tears or detachment. Secondary hemorrhage (vitreous or retinal) may also occur due to vascular shearing, as the vitreous body, having been displaced, may pull on the retina and contribute to further retinal pathology.”

Comment 2. Being consistence with the following terms of for example "spring-back," "rebound," "contrecoup," and "compression phase." Some sections use these interchangeably, that is unclear for readers who unfamiliar with ocular biomechanics.

Response to comment 2: In response to the reviewer’s comment, we have replaced the term “spring-back” with “rebound” to maintain consistency throughout the manuscript. The term “coup–contrecoup,” however, refers to rapid acceleration–deceleration forces within intraocular structures—particularly the vitreous—and is distinct from “rebound,” which describes the release of elastic energy from the tissues.

Comment 3. Stress values (e.g., 0.35 MPa, 0.37 MPa) are mentioned, mention the standard deviations if there are multiple simulations, or clarify if these are single-simulation results. Also, clarify if whether these stress levels exceed a specific value for the ocular tissues.

Response to comment 3: A single simulation was carried out under a worst-case scenario, assuming a direct, head-on impact from the pickleball without any angulation. This was further clarified in the Material at Methods:

Line 182: A single impact simulation was conducted, representing a direct strike to the center of the cornea by a pickleball traveling at 40 miles per hour, which corresponds to the average serve speed in a casual game”

Regarding to the stress levels, there’s a lack of failure characterization of intraocular tissues, given its intricate access for in-vivo testing, and delicate handling for in-vitro testing. The extraocular tissues like the sclera and cornea are more commonly analyzed, however, there are not the primary tissues at risk of failure during a pickleball impact (globe rupture is unfeasible).

Comment 4: The paper demonstrated stress and pressure peaks over time. It may enhance clarity to explicitly specify time points or normalized time units when these peaks occur.

Response to comment 4: Since this study is based on a single dynamic simulation, the use of normalized time was not applicable in this context. Instead, absolute time was used to report stress and pressure peaks as they naturally occurred within the simulation timeframe. For clarity, the specific time points at which these peaks occurred have been added throughout the manuscript, as recommended by the reviewer.

Comment 5: Add more detailed discussion on the modeling assumptions, particularly regarding: Whether vitreous liquefaction was considered, whether zonular fiber anisotropy was modeled, Simplifications at the vitreoretinal interface (no adhesion considered).). These assumptions will clarify the interpretation of the results.

Response to comment 5: Thank for the suggestion. Aged-related breakdown of the vitreous such as liquefaction was not considered in the simulation. As mentioned in line 203, humor bodies were modeled with properties similar to water, with a viscous component to account for shear resistance coming from inner collagen fibrils. This assumption resembles healthy vitreal conditions. Zonules were modeled as continuous material sheets. Although anatomically composed of thousands of discrete fibers, this approach captures their primary function of transmitting loads between the ciliary body and the lens, which shouldn’t compromise the biofidelity of the model. The following sentence was added to clarify the zonular modeling:

Line 129: “Zonules were modeled as a continuous material sheet, to capture the load transferring functions of discrete fiber arrangement.”

Line 189: “Due to modeling limitations at the ELC interface, vitreoretinal adhesion was assumed to be absent.”

Comment 6: Aging effects are mentioned briefly. Consider adding a short paragraph discussing how age-related changes in ocular biomechanic, which might influence the injury patterns observed in pickleball trauma.

Response to comment 6: Thank you for this suggestion. In response to your comment we have expanded the discussion and added more detailed description of age related changes in biomechanics of the eye.

Line 420: “Previous biomechanical studies have demonstrated that aging significantly alters the mechanical properties of the eye. With advancing age, the crystalline lens becomes progressively stiffer due to protein aggregation and a reduction in elasticity.35 In fact, lenses in older individuals can be up to four times stiffer compared to those in younger individuals.35 Computational models have shown that increased lens stiffness leads to elevated peak stress in the ciliary body, thereby raising the risk of tissue rupture during blunt ocular trauma. Similarly, corneal stiffness increases with age and is associated with elevated intraocular pressure and a diminished capacity to withstand deformation, which heightens the risk of corneal abrasion and rupture upon impact.36 Furthermore, age-related vitreous liquefaction reduces the eye’s ability to absorb mechanical shock, increasing the susceptibility to retinal detachment following trauma.37

Comment 7: Some figures (e.g., Fig. 2D, 3A, 5A) would benefit from clearer labeling of peaks and events (impact, rebound, etc.).

Response to comment 7: Thank you for the suggestion. The labeled peaks in Figs. 2D, 3A, and 5A correspond to local or global maxima in stress or pressure, but they are not tied to a single discrete event such as impact or rebound. These peaks emerge as mechanical waves propagate and reflect throughout the eye following the initial impact. Nonetheless, to improve interpretability, we have divided the time axis into impact and rebound phases, so the reader can now at which time point of the simulation the peaks appears.

Comment 8: Please include scale bars and color legends in all stress and pressure distribution figures for better interpretability.

Response to comment 8: All stress and pressure distribution figures already contain scale bars and color legends to support interpretation. If the concern pertains to Fig. 4, please note that panels A and B, as well as C and D, share the same scale bar, which is positioned on the left side of each respective row.

Comment 9: Where possible, draw stronger connections between simulation findings, and specific clinical case reports cited. this would strengthen the clinical relevance of the modeling results.

Response to comment 9: We appreciate the reviewer's insightful suggestion to enhance the clinical relevance of our findings. Below, we have revised the Discussion section (line 371) to explicitly link our simulation results with specific clinical case reports: 

Line 392: “Our finite element analyses align with previously reported clinical cases, providing biomechanical insight into the mechanisms underlying the diverse ocular injuries ob-served in pickleball-related trauma. Interestingly, our FEM identified peak zonular stresses (up to 0.37 MPa) during the rebound phase, which aligns with clinical reports of traumatic lens subluxation by Huang and Greven (2024). Their cases documented zonular fiber tears in patients after pickleball impacts, corroborating our finding that equatorial zonules are most vulnerable to dynamic forces. The observed shear stress concentrations at the ora serrata and posterior pole (Fig. 4A–B) correlate with retinal tears reported by Atkinson et al. (2022). Their cases high-lighted posterior pole tears following blunt trauma, consistent with our simulation’s prediction of elevated intraocular pressure (IOP) gradients disrupting retinal adhesion. Stress peaks in the iridocorneal angle (0.35 MPa, Fig. 2D) correspond to clinical observations of hyphema and angle recession by Waisberg et al. (2023). These injuries result from trabecular meshwork disruption, which our model attributes to circumfer-ential tensile stresses during limbal expansion. Asymmetric vitreous pressure gradients (Fig. 6A–C) explain cases of vitreous hemorrhage and posterior vitreous detachment in Boopathiraj et al. (2024), where tem-poral traction forces exceeded retinal resilience. These explicit connections reinforce the clinical validity of our FEM and its utility in explaining injury mechanisms observed in practice.”

Comment: Finite element material models may not fully capture viscoelastic behavior of ocular tissues. Discuss that carefully.

Response: Thank you for your comment. We have added a paragraph discussing the limitations of modeling viscoelastic behavior using finite element methods (FEM).

Line 519  “Moreover, while valuable for simulating ocular biomechanics, finite element models they may not fully capture the viscoelastic complexity of ocular tissues due to inherent simplifications in both geometry and constitutive behavior.48 Geometric approximations often diverge from the true anatomical features of the cornea and sclera.49,50 The accuracy and predictive power of these models depend on achieving an optimal balance between computational tractability and faithful representation of tissue microstructure and viscoelastic relaxation dynamics.51,52

Comment 12: Minor grammatical corrections needed throughout the text (e.g., "increase in participation" → "an increase in participation"; "potential of a tear" → "potential for a tear").

Response to comment 12: Thank you for this comment. Grammatical corrections have been introduced.

Comment 13. Consistently use "impact" instead of "hit" or "strike" unless describing an informal event.

Response to comment 13: Thank you for this valuable comment. In response to the reviewer’s comment, we have replaced “hit” in line 246 with “impact”

Comment 14: Show the impact of your research as future work. 

Response to comment 14: We have expanded the Conclusion and Future Work sections to outline translational applications and next steps.

Line 540: “Future studies will integrate polycarbonate lenses into our FEM to quantify stress re-duction in ocular structures. This will inform ASTM/ISO standards for pickleball-specific eyewear, addressing the lack of regulatory guidelines noted by The American Academy of Ophthalmology (2023). Moreover, to specifically study risks in aging population a parametric study will model age-related changes (e.g., scleral stiffening, vitreous syneresis) to evaluate their compounding effects on injury risk. This is critical given that 20% of players are >65 years old, a cohort with heightened susceptibility to retinal detachment and zonular weakness.”

Reviewer 2 Report

Comments and Suggestions for Authors

The manuscript entitled Finite Element Analysis of Ocular Impact Forces and potential 
complications in Pickleball-Related Eye Injuries was presented for the peer review by Cecary Rydz and coauthors.  My several issues are dealing with application of contact lenses by pickleball users and trauma associated with them. Have you considered it or not? Second, have you analyzed impact of time of the day and season on ocular injuries.

Model proposed by authors is adequate and complete for all type of injuries. Authors undertake a lot of interesting simulations.

Author Response

We sincerely thank the reviewer for taking the time to carefully read and evaluate our manuscript. Your thoughtful and constructive comments have greatly contributed to improving the clarity and quality of our paper on pickleball-related ocular injuries. 

Comment 1: The manuscript entitled Finite Element Analysis of Ocular Impact Forces and potential 
complications in Pickleball-Related Eye Injuries was presented for the peer review by Cezary Rydz and coauthors.  My several issues are dealing with application of contact lenses by pickleball users and trauma associated with them. Have you considered it or not? Second, have you analyzed impact of time of the day and season on ocular injuries.

Response:  Impact of Contact Lens Wear on Pickleball-Related Ocular Injuries.

In our paper, we did not specifically investigate the impact of contact lens wear on pickleball-related ocular injuries. However, previous reports on contact lens use in other sports provide useful insights into their potential protective benefits and limitations. Notably, a 2022 biomechanical study using porcine eyes demonstrated that soft contact lenses could withstand at least five times the force required to cause corneal abrasions compared to unprotected eyes.1 It is important to emphasize that the protective effect of soft contact lenses in this study was limited to preventing corneal abrasions.

In contrast, potential pickleball-related ocular injuries include hyphema, angle recession, lens subluxation, retinal hemorrhage, retinal tears, and retinal detachment—types of trauma against which soft contact lenses offer no protection.2–5 Furthermore, other studies have indicated that rigid contact lenses may actually increase the risk of ocular injury by shattering upon impact.6,7

Given these considerations, there is a critical need to further investigate the impact of contact lens wear on ocular injuries. This will be a subject of future research within our group, especially in light of the growing popularity of pickleball among older adults, who may already have pre-existing ocular vulnerabilities.

Comment 2: Model proposed by authors is adequate and complete for all type of injuries. Authors undertake a lot of interesting simulations.

Response: Impact of time of the day and season on ocular injuries.

While we have not yet investigated the impact of time of day and season on the incidence of pickleball-induced eye trauma, previous studies have demonstrated a seasonal pattern in sports-related ocular injuries. Notably, a 10-year study conducted in the U.S. reported that such injuries occurred most frequently during the summer months.8 This trend is likely due to increased participation in outdoor sports such as baseball, basketball, and soccer during this time (June to August in the Northern Hemisphere). Given the rising popularity of pickleball—especially as a popular outdoor sport played in similar seasonal conditions—we hypothesize that a comparable pattern may emerge. However, this remains to be systematically investigated. Additionally, few sports-related eye injuries have been reported at night.

References:

  1. Hou A, Jin ML, Goldman D. The Protective Effects of Soft Contact Lenses for Contact Sports: A Novel Porcine Model for Corneal Abrasion Biomechanics. Eye Contact Lens. 2022;48(5):228-230. doi:10.1097/ICL.0000000000000894
  2. Huang H, Greven MA. TRAUMATIC LENS SUBLUXATION FROM PICKLEBALL INJURY: A CASE SERIES. Retin Cases Brief Rep. 2024;18(1):15-17. doi:10.1097/ICB.0000000000001312
  3. Dang V, (Clinical MAC, 2021 undefined. Pickleball associated abrasion and iritis: a case study. clinicaloptometry.scholasticahq.comVT Dang, M AlkawallyCRO (Clinical & Refractive Optometry) Journal, 2021•clinicaloptometry.scholasticahq.com. Accessed August 13, 2024. https://clinicaloptometry.scholasticahq.com/article/36828.pdf
  4. Atkinson CF, Patron ME, Joondeph BC. RETINAL TEARS DUE to PICKLEBALL INJURY. Retin Cases Brief Rep. 2022;16(3):312-313. doi:10.1097/ICB.0000000000000965
  5. Boopathiraj N, Wagner I V., Krambeer CJ, et al. In a pickle: Cases of pickleball related ocular injuries. Am J Ophthalmol Case Rep. 2024;35. doi:10.1016/J.AJOC.2024.102082
  6. Mishra A, Verma AK. Sports related ocular injuries. Med J Armed Forces India. 2012;68(3):260. doi:10.1016/J.MJAFI.2011.12.004
  7. Micieli JA, Easterbrook M. Eye and Orbital Injuries in Sports. Clin Sports Med. 2017;36(2):299-314. doi:10.1016/J.CSM.2016.11.006
  8. Patel V, Pakravan P, Mehra D, Watane A, Yannuzzi NA, Sridhar J. Trends in Sports-Related Ocular Trauma in United States Emergency Departments from 2010 to 2019: Multi-Center Cross-Sectional Study. Semin Ophthalmol. 2023;38(4):333-337. doi:10.1080/08820538.2022.2107400